# Unifying Vision-Language Latents for Zero-label Image Caption Enhancement

**Sanghyun Byun**[*], **Jung Guack**[*], **Mohanad Odema**,
**Baisub Lee**, **Jacob Song**, **Woo Seong Chung**
LG Electronics USA
Santa Clara, CA 95054
{sang.byun, jung.guack, mohanad.odema}@lge.com
{baisub.lee, jaigak.song, wooseong.chung}@lge.com

**Editors:** Marco Fumero, Clementine Domine, Zorah Lähner, Irene Cannistraci, Bo Zhao, Alex Williams

## Abstract

Vision-language models (VLMs) achieve remarkable performance through large-scale image–text pretraining. However, their reliance on labeled image datasets limits scalability and leaves vast amounts of unlabeled image data underutilized. To address this, we propose *Unified **Vi**sion-Language Alignment for **Ze**ro-Label Enhancement (ViZer)*, an enhancement training framework that enables zero-label learning in image captioning, providing a practical starting point for broader zero-label adaptation in vision-language tasks. Unlike prior approaches that rely on human or synthetically annotated datasets, ViZer actively aligns vision and language representation features during training, enabling existing VLMs to generate improved captions without requiring text labels or full retraining. We demonstrate ViZer's advantage in qualitative evaluation, as automated caption metrics such as CIDEr and BERTScore often penalize details that are absent in reference captions. Applying ViZer on SmolVLM-Base and Qwen2-VL, we observe consistent qualitative improvements, producing captions that are more grounded and descriptive than their baseline.

## 1 Introduction

The integration of vision and language has enabled significant advances in multimodal AI, powering systems that can retrieve images, describe visual scenes, and reason across modalities. However, current vision-language models (VLMs) remain fundamentally limited by their reliance on labeled image–text datasets, which constrains scalability and leaves vast amounts of unlabeled image data underutilized. This dependence on scarce annotations not only restricts the scope of training but also contributes to persistent mismatches between visual encoders and language models, often manifesting as hallucinations, factually incorrect captions, and inconsistent multimodal reasoning, even in state-of-the-art systems. In this work, we focus on image captioning as our primary objective, since it is one of the simplest downstream tasks for exploring zero-label learning and provides a clear testbed for evaluating grounding and descriptive ability.

---

[*]Equal Contribution

Proceedings of the III edition of the Workshop on Unifying Representations in Neural Models (UniReps 2025).

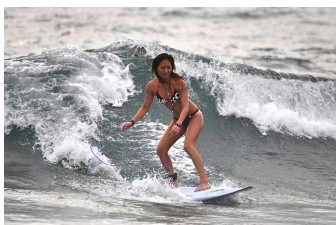 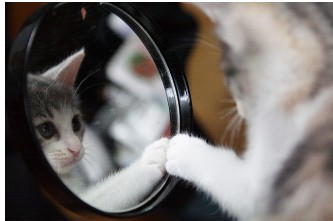 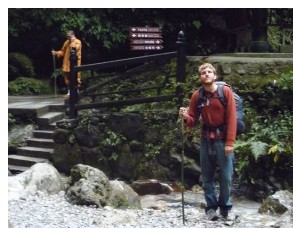

**GT:** A woman in a bikini riding a wave on a surfboard
**Base:** <PERSON> in 2008
**RL:** <PERSON> in 2008
**ViZer$_{GT}$:** Woman surfing in the ocean
**ViZer$_{G}$:** Surfer in a bikini riding a wave

**GT:** A cat playing with its reflection in a mirror
**Base:** My cat is so cute
**RL:** My cat has a mirror fetish
**ViZer$_{GT}$:** A cat looking at herself in a mirror
**ViZer$_{G}$:** A cat's paw in a mirror

**GT:** A man holding a stick standing next to a green hillside.
**Base:** <PERSON> at the start of the trail
**RL:** <PERSON> and <PERSON> on the trail
**ViZer$_{GT}$:** A man with a backpack and trekking poles on the trail
**ViZer$_{G}$:** A man crosses a stream on the way to the temple

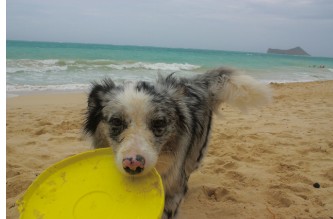 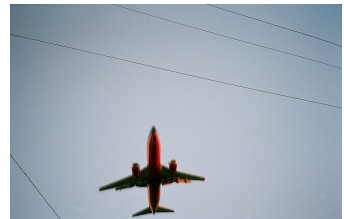 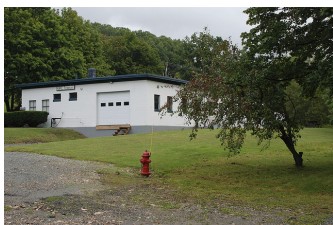

**GT:** There is a dog holding a Frisbee in its mouth
**Base:** My doggo loves the beach
**RL:** My doggo loves the beach
**ViZer$_{GT}$:** My dog playing fetch on the beach
**ViZer$_{G}$:** A pupper on the beach

**GT:** A colorful plane flying over head and telephone wires
**Base:** ITAP of an airplane
**RL:** ITAP of an airplane
**ViZer$_{GT}$:** ITAP of an airplane flying in the sky
**ViZer$_{G}$:** ITAP of an airplane flying over power lines

**GT:** A red fire hydrant sitting in a field next to a white building
**Base:** The firehouse in 2014
**RL:** The firehouse in 2014
**ViZer$_{GT}$:** The fire station in 2009
**ViZer$_{G}$:** The fire station in 2013

Figure 1: Qualitative captioning comparison for sample images from OpenImage dataset. We compare ground-truth, baseline, reinforcement-learning, ViZer$_{GT}$, and ViZer$_{G}$ generated captions. Information that include too much assumption or provide opinionated descriptions are colored in red. Added descriptions by incorporating ViZer or RL are colored in blue.

Recent advances in representation learning highlight the importance of predictive, latent-space modeling for semantic understanding. Frameworks like JEPA [18, 2] and DINO [7, 30] demonstrate that learning feature-level representations without relying on pixel-level reconstruction or dense supervision leads to more robust and generalizable features while improving data availability. However, these methods are primarily designed for visual representation learning, rather than directly generating grounded captions or synchronizing cross-modal semantics. As a result, they leave open the question of how to leverage joint-embedding alignment to improve downstream multimodal tasks, particularly when labeled caption data is scarce.

To address this challenge, we propose *Unified **Vi**sion-language Alignment for **Zer**o-label Enhancement (ViZer)*, a framework that enables enhancement with zero-label image caption training by actively aligning vision and language representations in latent space. Unlike traditional captioning models, which depend on annotated datasets, ViZer learns from raw images, predicting and synchronizing visual and linguistic semantics without requiring explicit labels. This makes ViZer fundamentally different from prior self-supervised methods. Instead of focusing solely on learning transferable features, it directly optimizes cross-modal alignment for the task of generative image captioning. Our contributions are threefold:

- **Active Latent Alignment**: We introduce an alignment mapper inspired by joint-embedding principles that learns bidirectional mappings between vision and language embeddings. Unlike static projection layers, ViZer continuously refines the alignment during training, enabling better semantic synchronization.

- **Zero-label Caption Training**: ViZer introduces a novel training paradigm where captions are learned with unlabeled images. By enforcing consistency between

predicted visual representations and generated captions, the model self-improves using only raw images.

- **Post-hoc Integration**: Any VLM architectures utilizing a vision encoder can be trained with ViZer. Utilizing a lightweight semantic bridge, it enhances alignment while adding minimal computational overhead.

## 2 Related Works

### 2.1 Vision-Language Model Training Paradigms

The evolution of vision-language models has followed several distinct paradigms, each addressing different aspects of multimodal understanding. Early approaches focused on task-specific architectures, but the field has increasingly moved towards unified, general-purpose models.

**Contrastive Learning Approaches**: CLIP [32] pioneered large-scale contrastive training between images and texts, learning a shared embedding space by maximizing similarity between matched pairs while minimizing it for mismatched ones. ALIGN [15] scaled this approach to billions of noisy image-text pairs, demonstrating robustness to data quality. SigLIP [46] improved training efficiency through sigmoid loss formulation, eliminating the need for global batch statistics. These models excel at zero-shot transfer but produce static alignments that are not continuously aligned with output text in training.

**Generative Pretraining**: Models like SimVLM [40] and BLIP [19] combine vision encoders with language models for image-conditioned text generation. BLIP-2 [20] introduced Q-Former, a lightweight module that bridges frozen image encoders and LLMs through learnable queries. However, these approaches still rely on limited captioning datasets for alignment and struggle to generalize beyond their training distribution.

**Multimodal LLMs**: Recent works integrate visual encoders directly with large language models. LLaVA [26] uses simple linear projection to connect CLIP encoders with LLMs, while Flamingo [1] employs cross-attention layers for deeper integration. Despite their impressive capabilities, these models inherit the alignment limitations of their frozen vision components, leading to persistent hallucination issues [12, 23].

### 2.2 Image-Text Alignment Techniques

The challenge of aligning visual and textual representations has been approached from multiple angles, ranging from architectural innovations to training objectives.

**Cross-Modal Attention Mechanisms**: ViLBERT [27] introduced co-attentional transformer layers that process vision and language streams separately before fusing them. LXMERT [35] extended this with specialized cross-modality encoders. More recently, models like Unified-IO 2 [28] demonstrate that unified transformer backbones can process multiple modalities through shared attention mechanisms. However, these approaches require training from scratch and cannot leverage existing pretrained components.

**Adaptive Projection Layers**: Several works attempt to learn better mappings between frozen encoders. Frozen [37] trains a vision encoder to produce embeddings in the language model's input space. MAGMA [10] and Fromage [16] use learnable adapters to align pretrained vision and language models. While more flexible than static projections, these methods still operate on fixed representations and cannot dynamically adjust alignment based on context.

**Multimodal Fusion Strategies**: Recent architectures explore different fusion mechanisms. UniFork [22] proposes a Y-shaped architecture that shares shallow layers while maintaining task-specific branches. Models like Qwen2-VL [42] and Janus-Pro [8] introduce sophisticated fusion modules that balance shared learning with specialization. Yet these approaches focus on architectural design rather than addressing the fundamental alignment problem.

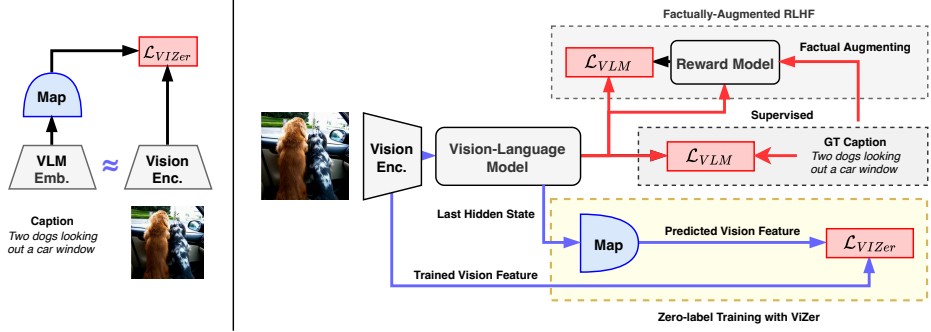

Figure 2: Unifying vision-language feature alignment with ViZer for zero-label image caption enhancement. (Left) Given text and vision features extracted through VLM modules, we train a mapper that maximizes the cosine similarity between vision features and mapped text features. (Right) Compared to existing vision-language training, ViZer does not require any textual guidance.

## 2.3 Unified Representations and Label-Efficient Learning

The pursuit of unified representations that capture semantic structure across modalities has led to significant advances in predictive learning frameworks, many of which aim to reduce reliance on human-annotated labels.

**Joint Embedding Predictive Architectures (JEPA)**: The JEPA framework [18] advocates for learning through prediction in abstract representation space rather than pixel space. I-JEPA [2] applies this to images, predicting representations of masked regions from visible context. This approach learns more semantic features without relying on handcrafted augmentations. V-JEPA [6] extends this to video, demonstrating that temporal prediction in latent space captures motion and object permanence. MC-JEPA [5] unifies motion and content learning, showing that joint objectives benefit both tasks.

**Self-Distillation Methods**: DINO [7] reveals that Vision Transformers can develop emergent semantic properties through self-distillation, including explicit segmentation without human labels. DINOv2 [30] scales this approach to larger models and datasets, producing features that excel across diverse downstream tasks without extensive fine-tuning.

**Cross-Modal Predictive Learning**: Beyond unimodal approaches, recent works explore predictive objectives that leverage multi-view or paired data to discover cross-modal correspondences. Data2vec [4] learns unified representations for speech, vision, and text through masked prediction. ImageBind [11] aligns six modalities using images as a binding mechanism. While these techniques reduce reliance on curated labels, they are often evaluated in settings where paired captions are still present.

## 3 Method

Vizer proposes to leverage the potential of joint embeddings to optimize the alignment of image and text directly on the downstream task. In this paper, we first test this hypothesis on zero-label image caption training without annotations. Using joint-embedding objectives, it actively aligns image and language representations for captioning, further enhancing VLM performance. ViZer bidirectionally aligns entire representation spaces, allowing multimodal models to self-improve using only raw images.

### 3.1 Unified Vision-language Alignment

Vision-language models (VLMs) such as CLIP or LIFT rely on early-stage image-text alignment to connect modalities. However, this alignment is often performed statically during pretraining and does not extend to downstream integration with foundational models, such as large language models (LLMs). This leaves a fundamental gap: while the VLM and

LLM may be individually powerful, their latent spaces are not directly co-adapted, resulting in possible representational mismatch in multimodal tasks.

ViZer is designed to bridge the representational gap between visual and textual modalities explicitly. We first introduce a mapper that focuses on directly aligning the latent features produced by the vision encoder with the token-level embeddings of the LLM, enabling seamless fusion between vision and language that is later used to enhance VLM performance.

**Architecture**   Given a vision-language model with a frozen vision encoder $V_\theta$ and text encoder $E_\phi$ (tokenizer and embedding function), we define the ViZer mapping function as

$$M_\tau(\cdot) = h_\tau(f_\psi(\cdot))$$

where $h$ is a multi-layer perceptron (MLP) that transforms textual embeddings to matched visual features, and $f_\psi$ is the VLM transformer layers without the LM head. Specifically, visual features $F_I = V_\theta(I)$ are extracted from an image $I$, and textual features are computed as $\hat{F}_T = M_\tau(E_\phi(x_{:t}))$ where $t$ is the length of caption tokens. The ViZer mapper $M_\tau$ is parameter-efficient and can scale with dataset size or model size, making it ideal for modular adaptation.

**Training Objective**   ViZer is trained using a contrastive loss that encourages alignment between visual features and the mapped text representations. Specifically, we minimize the cosine distance between the two modalities following previous works [36]:

$$\mathcal{L}_{ViZer} = 1 - \frac{F_I \cdot \hat{F}_T}{\|F_I\| \times \|\hat{F}_T\|}$$

This formulation promotes directional and semantic similarity in the shared latent space, enabling effective multimodal grounding.

**Training ViZer mapper on Image-Only Data**   We introduce two variants of the mapper. ViZer$_{\text{GT}}$ mapper is trained on ground-truth image-text pairs, while ViZer$_{\text{G}}$ mapper is aligned on the hidden features of VLM to be trained. In both versions of ViZer, the VLM training portion (Section 3.2) does not reference any text, making it effectively unsupervised. However, considering that the ground-truth caption in the mapper can affect the VLM output, only ViZer$_{\text{G}}$ may be considered truly unsupervised and zero-label. Thus, we define ViZer$_{\text{G}}$ as:

$$\hat{F}_T = M_\tau(f_\psi(V_\theta(I) \circ E_\phi(P))_{t+1:})$$

where $P$ is a textual prompt for image captioning, and the output of $f_\psi(\cdot)_{t+1:}$ is the generated caption from the VLM trained from image and prompt tokens of combined length $t$. This first projects the ground-truth captions to VLM hidden features for alignment with vision features.

As low-quality generated captions can result in poor alignment in mapper training, it is crucial that we feed images that the model has already seen during previous training steps. A straightforward approach is to utilize commonly used datasets, such as COCO [25] and CC3M [33], for training.

## 3.2   Zero-label Learning for Image Captioning

While recent advances in supervised and reinforcement learning approaches [34, 1] have improved the quality of vision tasks, these methods remain constrained by their reliance on curated image–text pairs. Such dependence not only limits scalability to internet-scale data but also introduces stylistic and semantic biases that arise from the annotation process or reward design, reducing the model's ability to generalize to open-world or long-tail scenarios. To improve data scalability, we utilize the introduced ViZer mappers to train the VLMs directly on unlabeled images.

Table 1: Image captioning performance of VLMs on automatic evaluation metrics. Although a comparison between RL and ViZer is not fair as RL has access to image captions, we compare them regardless. Results are reported on F1 scores for all. $\text{ViZer}_{GT}$ and $\text{ViZer}_{G}$ denote ViZer mappers trained with ground truth labels and generated labels, respectively. Models trained and evaluated are SmolVLM-Base (2.25B) [29] and Qwen2-VL-2B-Instruct [39]. The best metrics for each model are highlighted in green.

| | | COCO | | | | | | | | CC3M | | | | | | | |
| | | $\text{BLEU}_1$ | $\text{BLEU}_2$ | $\text{BLEU}_3$ | $\text{BLEU}_4$ | BERTS | $\text{ROUGE}_L$ | CIDEr | CLIPS | $\text{BLEU}_1$ | $\text{BLEU}_2$ | $\text{BLEU}_3$ | $\text{BLEU}_4$ | BERTS | $\text{ROUGE}_L$ | CIDEr | CLIPS |
|---|---|---|---|---|---|---|---|---|---|---|---|---|---|---|---|---|---|
| Base | Smol | 0.3784 | 0.2424 | 0.1580 | 0.1053 | 0.7187 | 0.2712 | 0.2760 | 0.2529 | 0.2276 | 0.1295 | 0.0847 | 0.0604 | 0.6735 | 0.2093 | 0.5601 | 0.2617 |
| | Qwen | 0.5249 | 0.3646 | 0.2512 | 0.1749 | 0.8077 | 0.3817 | 0.5205 | 0.2693 | 0.1833 | 0.0920 | 0.0512 | 0.0310 | 0.6668 | 0.1685 | 0.3394 | 0.2766 |
| RL | Smol | 0.3623 | 0.2276 | 0.1457 | 0.0956 | 0.7093 | 0.2657 | 0.2549 | 0.2506 | 0.2187 | 0.1206 | 0.0761 | 0.0521 | 0.6682 | 0.2032 | 0.5239 | 0.2604 |
| | Qwen | 0.4910 | 0.3350 | 0.2253 | 0.1535 | 0.8015 | 0.3660 | 0.4819 | 0.2689 | 0.1821 | 0.0883 | 0.0469 | 0.0270 | 0.6655 | 0.1624 | 0.3248 | 0.2762 |
| $\text{ViZer}_{GT}$ | Smol | 0.5564 | 0.3763 | 0.2486 | 0.1624 | 0.7826 | 0.3826 | 0.5052 | 0.2569 | 0.2287 | 0.1253 | 0.0773 | 0.0517 | 0.6718 | 0.2111 | 0.5439 | 0.2647 |
| | Qwen | 0.4937 | 0.3240 | 0.2084 | 0.1358 | 0.7953 | 0.3526 | 0.4621 | 0.2704 | 0.1846 | 0.0937 | 0.0533 | 0.0329 | 0.6683 | 0.1704 | 0.3501 | 0.2769 |
| $\text{ViZer}_{G}$ | Smol | 0.4081 | 0.2728 | 0.1824 | 0.1221 | 0.7455 | 0.3037 | 0.3374 | 0.2571 | 0.2303 | 0.1289 | 0.0824 | 0.0571 | 0.6756 | 0.2112 | 0.5514 | 0.2636 |
| | Qwen | 0.5373 | 0.3687 | 0.2484 | 0.1684 | 0.7990 | 0.3800 | 0.4697 | 0.2744 | 0.1773 | 0.0882 | 0.0487 | 0.0290 | 0.6641 | 0.1670 | 0.3263 | 0.2774 |

**Training Approach** Our method targets smaller VLMs, which are more prone to underfitting and harder to optimize with limited supervision. Instead of relying on hand-labeled captions, we leverage the aligned latent representations produced by $\text{ViZer}_{GT}$ and $\text{ViZer}_{G}$ to guide caption generation. Specifically, we treat the generated caption as a hypothesis and measure its agreement with the image via the aligned embedding space.

The model is trained to minimize semantic discrepancy between the caption-derived text embeddings and the corresponding visual features. To integrate this with the captioning model without disturbing its pretrained capabilities, we apply Low-Rank Adaptation (LoRA) to the VLM. This enables selective tuning during captioning while preserving zero-shot performance for other tasks by turning it off. To ensure consistent alignment, we use the same cosine similarity used to train ViZer for mapper: $\mathcal{L}_{zero} = \mathcal{L}_{ViZer}$.

## 4 Experiments

### 4.1 Implementation Details

We evaluate ViZer on two vision-language models with varying scales and architectures: SmolVLM-Base (2.25B) [29] and Qwen2-VL-2B-Instruct [39]. These models differ in pre-training corpus, vision backbones, and tokenization strategies, providing a diverse testbed for evaluating the generality and robustness of ViZer. All mappers are trained on a mixture of COCO [25] and CC3M [33], while the VLM is trained strictly on the non-labeled OpenImagesV7 [17] dataset.

We fine-tune the model with LoRA, configured with a rank set to 32, alpha to 64, and dropout to 0.1, without bias. ViZer mapper depth is fixed to 2, with varying MLP widths. Mapper and VLM are both trained for 1 epoch for all variants of ViZer. AdamW optimizer is used with a weight decay of 0.01. Pre-trained models are attained with Hugging Face [41]. We train all models on an RTX 4090 (24GB VRAM). All models are implemented in PyTorch. Prompt templates used for the reward model in reinforcement learning and those used for image captioning are included below. For a reinforcement learning comparison, we compare our results with those obtained using Qwen2.5-VL-3B-Instruct-AWQ [39] as the reward model, which helps save memory for training. We keep the LoRA configurations equal.

In all cases, we integrate the ViZer mapper between the frozen vision encoder and the VLM's hidden feature space, training it jointly with unified vision latent objectives as described in Section 3. We evaluate captions using standard metrics, including ROUGE-L [24], BLEU [31], CIDEr [38], BERTScore [47] (F1), and CLIPScore [13] and report mean improvements over the respective baselines along with qualitative comparisons.

Table 2: Image captioning performance of VLMs with varying ViZer mapper width and mapping data size on automatic evaluation metrics. Results are reported on F1 scores for all. $ViZer_{GT}$ and $ViZer_G$ stand for ViZer mapper trained with ground truth labels and generated labels, respectively. The model trained and evaluated is SmolVLM-Base (2.25B).

| | size | width | COCO | | | | | | | CC3M | | | | | | |
|---|---|---|---|---|---|---|---|---|---|---|---|---|---|---|---|---|
| | | | $BLEU_1$ | $BLEU_2$ | $BLEU_3$ | $BLEU_4$ | BERTS | $ROUGE_L$ | CIDEr | $BLEU_1$ | $BLEU_2$ | $BLEU_3$ | $BLEU_4$ | BERTS | $ROUGE_L$ | CIDEr |
| $ViZer_{GT}$ | 10k | 128 | 0.4077 | 0.2738 | 0.1852 | 0.1263 | 0.7428 | 0.3058 | 0.3407 | 0.2176 | 0.1206 | 0.0764 | 0.0524 | 0.6738 | 0.2030 | 0.5252 |
| | | 256 | 0.4064 | 0.2739 | 0.1859 | 0.1279 | 0.7404 | 0.3036 | 0.3415 | 0.2186 | 0.1214 | 0.0772 | 0.0534 | 0.6740 | 0.2044 | 0.5322 |
| | | 512 | 0.3802 | 0.2545 | 0.1716 | 0.1165 | 0.7356 | 0.2924 | 0.3164 | 0.2177 | 0.1213 | 0.0775 | 0.0534 | 0.6741 | 0.2055 | 0.5309 |
| | | 1024 | 0.3853 | 0.2593 | 0.1750 | 0.1200 | 0.7399 | 0.3001 | 0.3293 | 0.2155 | 0.1207 | 0.0773 | 0.0537 | 0.6743 | 0.2046 | 0.5334 |
| | 40k | 128 | 0.3928 | 0.2620 | 0.1758 | 0.1196 | 0.7347 | 0.2942 | 0.3244 | 0.2217 | 0.1243 | 0.0794 | 0.0549 | 0.6741 | 0.2063 | 0.5414 |
| | | 256 | 0.4169 | 0.2827 | 0.1922 | 0.1326 | 0.7480 | 0.3113 | 0.3551 | 0.2213 | 0.1235 | 0.0790 | 0.0550 | 0.6748 | 0.2057 | 0.5361 |
| | | 512 | 0.4173 | 0.2816 | 0.1910 | 0.1307 | 0.7423 | 0.3074 | 0.3524 | 0.2237 | 0.1248 | 0.0798 | 0.0552 | 0.6738 | 0.2061 | 0.5434 |
| | | 1024 | 0.4110 | 0.2772 | 0.1873 | 0.1278 | 0.7421 | 0.3062 | 0.3512 | 0.2235 | 0.1250 | 0.0796 | 0.0549 | 0.6744 | 0.2074 | 0.5441 |
| | 100k | 128 | 0.3936 | 0.2601 | 0.1736 | 0.1177 | 0.7335 | 0.2915 | 0.3175 | 0.2218 | 0.1249 | 0.0806 | 0.0563 | 0.6744 | 0.2066 | 0.5476 |
| | | 256 | 0.4161 | 0.2823 | 0.1911 | 0.1304 | 0.7433 | 0.3084 | 0.3541 | 0.2231 | 0.1243 | 0.0792 | 0.0547 | 0.6739 | 0.2058 | 0.5385 |
| | | 512 | 0.3987 | 0.2654 | 0.1785 | 0.1213 | 0.7359 | 0.2969 | 0.3316 | 0.2227 | 0.1247 | 0.0798 | 0.0553 | 0.6745 | 0.2071 | 0.5431 |
| | | 1024 | 0.3937 | 0.2612 | 0.1755 | 0.1195 | 0.7342 | 0.2942 | 0.3196 | 0.2235 | 0.1258 | 0.0807 | 0.0563 | 0.6741 | 0.2078 | 0.5479 |
| $ViZer_G$ | 10k | 128 | 0.3850 | 0.2419 | 0.1540 | 0.0990 | 0.7334 | 0.2774 | 0.2745 | 0.2279 | 0.1267 | 0.0805 | 0.0557 | 0.6742 | 0.2103 | 0.5444 |
| | | 256 | 0.4112 | 0.2653 | 0.1722 | 0.1123 | 0.7381 | 0.2948 | 0.3133 | 0.2303 | 0.1289 | 0.0824 | 0.0571 | 0.6756 | 0.2112 | 0.5514 |
| | | 512 | 0.3707 | 0.2344 | 0.1500 | 0.0964 | 0.7275 | 0.2740 | 0.2682 | 0.2292 | 0.1290 | 0.0829 | 0.0575 | 0.6753 | 0.2123 | 0.5575 |
| | 40k | 128 | 0.3735 | 0.2357 | 0.1513 | 0.0972 | 0.7223 | 0.2703 | 0.2657 | 0.2259 | 0.1261 | 0.0800 | 0.0551 | 0.6747 | 0.2096 | 0.5430 |
| | | 256 | 0.3662 | 0.2283 | 0.1453 | 0.0923 | 0.7197 | 0.2655 | 0.2542 | 0.2256 | 0.1262 | 0.0809 | 0.0563 | 0.6741 | 0.2097 | 0.5499 |
| | | 512 | 0.3630 | 0.2285 | 0.1472 | 0.0951 | 0.7201 | 0.2659 | 0.2567 | 0.2221 | 0.1237 | 0.0787 | 0.0547 | 0.6731 | 0.2066 | 0.5364 |
| | 100k | 128 | 0.3521 | 0.2196 | 0.1400 | 0.0901 | 0.7149 | 0.2587 | 0.2430 | 0.2241 | 0.1250 | 0.0800 | 0.0557 | 0.6735 | 0.2078 | 0.5393 |
| | | 256 | 0.3570 | 0.2254 | 0.1453 | 0.0944 | 0.7180 | 0.2639 | 0.2539 | 0.2239 | 0.1255 | 0.0801 | 0.0556 | 0.6737 | 0.2078 | 0.5431 |
| | | 512 | 0.3338 | 0.2066 | 0.1309 | 0.0833 | 0.7124 | 0.2493 | 0.2297 | 0.2213 | 0.1237 | 0.0792 | 0.0550 | 0.6734 | 0.2066 | 0.5387 |

## 4.2 Prompt Templates

### 4.2.1 Reward Model Prompt for Reinforcement Learning

We feed the following prompt to get a score in range of 1 to 0 (translated from A-E). We use the following chat template for reward score generation:

> </im_start/>assistant
> Score:
> </im_start/>system
> You are a helpful assistant …
> </im_start/>user
> Look at this image and evaluate the caption quality.
> Reference (human-written): *<GROUND-TRUTH CAPTION>*
> Generated caption: *<VLM-GENERATED CAPTION>*
> Rate the generated caption:
> A) Excellent - Perfect description, captures all key details
> B) Good - Accurate main objects/scene, minor missing details
> C) Fair - Correct general scene but lacks specificity
> D) Poor - Some correct elements but major inaccuracies
> E) Wrong - Completely incorrect or irrelevant
> Answer with just the letter (A, B, C, D, or E):

### 4.2.2 Image Caption Prompt

SmolVLM-Base does not require any prompting for image captioning. We use the following general format for prompting Qwen2-VL-2B-Instruct for image captioning:

> </im_start/>system
> You are a helpful assistant.
> </im_start/>user
> </image_start/>*<IMAGE>*</image_end/>
> Describe this image in the shortest form.
> </im_start/>assistant

## 4.3   Image Captioning

**Quantitative Results**   Table 1 presents results when the ViZer mapper is trained with ground-truth (ViZer$_{GT}$) or generated (ViZer$_{G}$) image-caption pairs before VLM training. We notice an overall increase in CLIPScore for both variants of ViZer, indicating likely improved image caption performance. Although we observe minimal differences across the metrics, with improvements particularly evident in COCO tests on SmolVLM, quantitative results are not the fairest point of comparison here, as captions can be subjective, and we only compare the generated text against the ground-truth caption. However, the generated caption may include more insightful details about the image than the ground-truth, especially since they were not trained on any of the ground-truth texts, particularly for ViZer$_{G}$.

Automated evaluation metrics commonly used in image captioning, such as BLEU [31], CIDEr [38], and ROUGE [24], are inherently limited because they compare generated captions to reference texts rather than evaluating consistency with the image itself. Since no single caption can fully represent all visual details, these metrics often undervalue captions that are semantically correct but phrased differently or include additional details from the references. Although reference-free methods [9, 21, 14] such as CLIPScore [13] and HICEScore [45] have been proposed, they also present limitations that hinder their practical use. We discuss this further in Section 5

**Qualitative Results**   To better capture performance, we place a stronger emphasis on qualitative comparisons, which directly reveal how our approach improves visual grounding and contextual accuracy. For our experiments, we select the best-performing settings from Table 2, using SmolVLM-Base [29] and Qwen2-VL [39]. We set the mapper width to 256 for all configurations, with 40k samples for ViZer$_{GT}$ and 10k samples for ViZer$_{G}$. We also experiment with a reinforcement learning (RL)-based variant trained on reward scores.

Figure 1 shows qualitative improvement over baseline and RL. Because the reward signals are inherently biased toward assigning higher scores to avoid destabilizing pretrained representations for reinforcement learning, the resulting captions remain close to the originals, with only marginal qualitative differences. In contrast, our zero-label framework, ViZer, which leverages self-generated captions, demonstrates several interesting behaviors. It frequently introduces additional scene-relevant attributes, such as "*surfing*" or "*mirror*," which enrich captions when visually accurate. We also observe that the model emphasizes prominent visual cues, such as "*power lines*," demonstrating its ability to capture strong structural features without explicit supervision. However, ViZer faces challenges when the baseline captions lack sufficient context or are incorrect. In such cases, improvements are limited, and the model often defaults to trivial edits, such as adjusting numerical details (e.g., years in the firehouse example). Despite this, the model remains robust and avoids catastrophic degradation, preserving overall caption quality even when enhancements are minimal.

Figure 5 and Figure 6 in the supplementary appendix present further qualitative comparisons of SmolVLM-Base [29] across four configurations: baseline, RL, ViZer$_{GT}$, and ViZer$_{G}$. Across a wide range of examples, our approach produces captions with stronger contextual grounding, fewer hallucinations, and better incorporation of subtle visual details compared to both the baseline and RL models. The generated captions more accurately describe objects, relationships, and textures while avoiding fabricated elements commonly present in baseline outputs. These qualitative findings align fairly with our quantitative results, where labeled-caption ViZer maintains consistent performance across ROUGE, BLEU, CIDEr, and BERTScore. Overall, our method produces captions that are more visually faithful, contextually grounded, and semantically descriptive, demonstrating its effectiveness as a scalable strategy for improving multimodal alignment.

## 4.4   ViZer Mapper Width and Sample Size

To better understand the trade-offs in aligning visual and textual representations, we investigate the impact of both ViZer mapper width and the size of its training dataset. The ViZer mapper plays a central role in bridging text and visual embeddings, so its representational capacity and data regime directly influence downstream captioning quality. Table 2 summarizes quantitative results across different configurations.

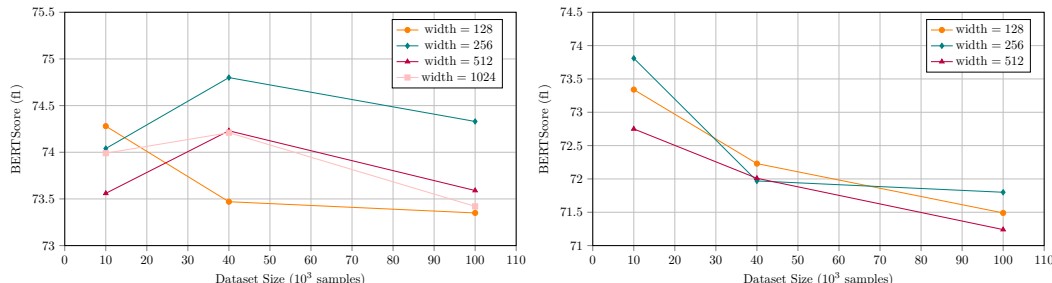

Figure 3: Visualization of ViZer mapper width to mapping data size comparison for SmolVLM-Base [29]. (right) $ViZer_{GT}$ and (left) $ViZer_G$ both display best results with a width of 256, with a preference for smaller dataset sizes.

**Preferred Dataset Sizes and Overfitting Effects.** Interestingly, we find that smaller training dataset sizes are consistently preferred for both $ViZer_{GT}$ and $ViZer_G$. Specifically, the best performance emerges with roughly 40k labeled samples for $ViZer_{GT}$ and around 10k samples for $ViZer_G$. Increasing the sample size beyond these ranges yields diminishing returns and, in several cases, even leads to measurable performance degradation. Figure 3 visualizes the effect of varying ViZer mapper width based on mapper dataset size. The empirical results display a pattern in which smaller-width MLPs perform better than larger, overfit networks.

This effect appears tied to overfitting. For $ViZer_{GT}$, training on larger ground-truth datasets encourages the mapper to memorize captioning patterns specific to the reference corpus, reducing its robustness when handling out-of-domain or noisy inputs. For $ViZer_G$, the risk is even more pronounced: self-generated captions often omit details or introduce noise, and excessive reliance on them pushes the mapper toward incomplete semantic alignment. By keeping training datasets relatively small, the mapper learns to maintain a looser but more flexible alignment in the visual feature space, which generalizes better to diverse scenarios. This observation highlights that bigger datasets are not always better when labels are scarce or noisy, and careful control of sample size can improve model robustness.

**Use of Captioning Accuracy as a Proxy.** Although Table 2 reports captioning accuracy across mapper widths and dataset sizes, we stress that this metric is only a proxy. Our objective is not to maximize captioning benchmarks per se but to identify settings that promote better cross-modal alignment. Captioning accuracy offers a practical way to compare configurations under controlled conditions, yet it does not directly measure improvements in downstream reasoning or descriptive grounding. Accordingly, these results should be interpreted as guidance for selecting mapper widths and training regimes, rather than as definitive indicators of ViZer's alignment quality. Future work may benefit from evaluation metrics that better capture visual grounding, semantic coverage, and generalization.

## 5 Discussion and Future Works

A central challenge lies in the inadequacy of automated evaluation metrics for image captioning. Metrics such as CIDEr [38], BLEU [31], and BERTScore [47] reduce evaluation to surface-level overlap with reference captions and often penalize outputs that correctly capture details missing from annotations, while reference-free metrics such as InfoMetIC [14] and CLIPScore [13] result in large overhead for sub-par evaluation improvements. This issue is pronounced for long-tail or complex images, where references frequently omit objects or attributes that ViZer-generated captions include. More reliable evaluation may require metrics that account for semantic coverage or visual grounding, for example, by comparing predicted captions against structured visual features or through targeted human assessments. For reference-free methods, CLIPScore [13] operates similarly to our ViZer mapper, using cosine similarity between visual and text features. However, as shown in Figure 4 in the supplementary appendix, ground-truth captions often do not align perfectly with image features, leading to unreliable scores. HICEScore [45] is designed for natural images with

segmentation-based evaluation, making it more suitable for object labeling than for diverse captioning tasks. Moreover, its performance on general caption generation remains unverified, and no official implementation is available, limiting reproducibility. Lastly, the performance of reference-free metrics, as reported by Hu et al. [14], clearly displays similar limitations to reference-based methods, failing to overcome reference-based evaluation on the ground-truth captions of Flickr8k [43]. It is crucial to note that point of reference-free image caption evaluation should aim to significantly improve upon reference-based methods, rather than achieving similar performance. While our qualitative analysis reveals that ViZer generates more grounded and descriptive captions, the lack of suitable metrics hinders the recognition of these gains. It makes fair comparison difficult with supervised baselines. These challenges highlight the need for a reference-free metric that is (1) image-native with object–interaction understanding, (2) openly available, and (3) aware of varying caption lengths and image complexities.

Another limitation is the uncertain behavior of ViZer on highly diverse or out-of-distribution images. Our results suggest benefits when training on images similar to those used in pretraining, but performance under substantially different distributions remains unclear. Domains with distinct visual statistics, such as medical or satellite imagery, or noisy web data, may challenge the robustness of zero-label alignment. Addressing this will likely require systematic studies of dataset diversity and domain adaptation strategies. These questions become particularly important if ViZer is to scale to real-world deployments where models inevitably encounter heterogeneous visual data.

Finally, ViZer demonstrates that zero-label enhancement training is a viable approach for leveraging unlabeled images to strengthen multimodal systems. Image captioning provides a clear and interpretable starting point, but the same principles can extend to tasks such as visual question answering, multimodal reasoning, or dialogue grounded in images. Similar to how self-supervised vision frameworks such as DINO [30] and V-JEPA [3] showed that large-scale, label-free pretraining can yield strong visual representations, ViZer highlights that analogous strategies are effective in the vision–language space. Scaling ViZer to internet-scale datasets could enable models to continually refine their alignment as new data becomes available, reducing reliance on annotation pipelines and moving closer to adaptive, label-free multimodal learning.

## 6 Conclusion

We introduced ViZer, a framework for enhancing zero-label image captioning that aligns vision and language representations without requiring ground-truth captions or full retraining. Applied to SmolVLM-Base and Qwen2-VL, ViZer consistently produces captions that are more grounded and descriptive than their baselines, demonstrating that unlabeled images can directly improve generative performance. While standard metrics such as CIDEr and BERTScore often fail to capture these gains by penalizing correct details absent from reference captions, qualitative evaluation highlights ViZer's ability to generate richer and more accurate descriptions. Beyond captioning, ViZer illustrates that zero-label adaptation is a practical strategy for enhancing existing vision-language models: it can be modularly integrated without disrupting unrelated tasks, scales to new architectures, and leverages unlabeled data as a training signal. Looking forward, captioning serves as a first step toward broader applications, such as visual question answering and multimodal reasoning. Scaling ViZer to internet-scale image collections may open a path toward fully label-free training, capable of improving grounding, descriptive accuracy, and adaptability across diverse domains.

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
