# A  Additional Qualitative Samples

Numerically, previous findings [44] have shown it is challenging to see the difference due to the nature of automated text scoring criteria. Thus, we include sample image-caption pairs showing the qualitative improvement in image caption as ViZer-trained VLMs display a more accurate description of an image with minimal garbage. We include SmolVLM-Base [29] in Figure 5 and Figure 6, and Qwen2-VL [39] results in Figure 7 and Figure 8.

## A.1  Vision Feature Space Visualizations

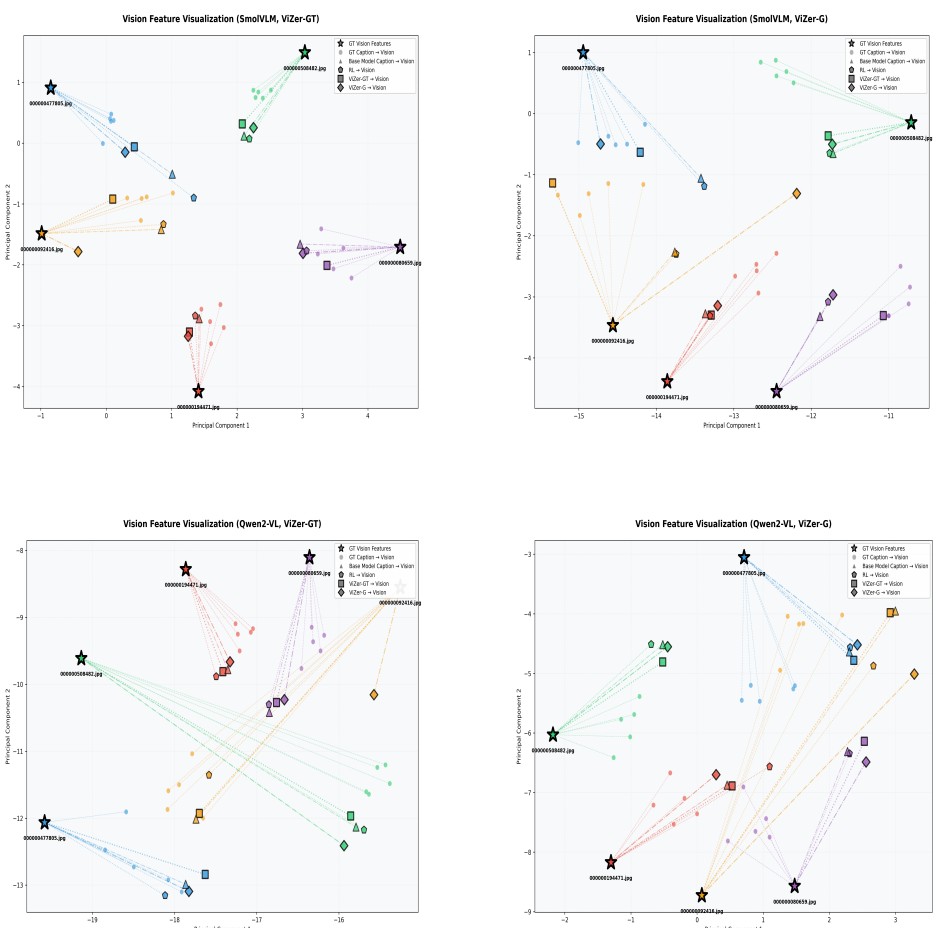

Figure 4: Visual Feature Comparisons on post-training generated image captions for SmolVLM-Base [29] (top) and Qwen2-VL [39] (bottom) with ViZer$_{GT}$ (left) and ViZer$_{G}$ (right). 2-dimensional PCA is applied to text and vision features for ease of comparison. Points closer to stars represent similar latents.

To better understand the impact of ViZer on vision-language models (VLMs), we visualize the 2-dimensional PCA projections of visual features for a sample image, its ground-truth captions, the original generated captions, and the post-ViZer generated captions in Figure 4. The visualization demonstrates that ViZer generally brings the generated caption embeddings closer to the corresponding visual features, indicating improved semantic alignment between modalities. However, the features are not perfectly overlapping, which reflects a key property of our approach: while ViZer enhances grounding, it avoids forcing captions to perfectly match visual embeddings. This balance is important because no textual description can fully capture all aspects of an image, and overly aggressive alignment may lead to unnecessarily verbose or unnatural captions.

# B  Evaluation Metrics

## B.1  BLEU

The BLEU [31] (Bilingual Evaluation Understudy) score measures the similarity between a generated sentence $\hat{y}$ and one or more reference sentences $y$. It is based on modified $n$-gram precision, combined using a weighted geometric mean and adjusted by a brevity penalty (BP) to avoid favoring shorter outputs. Formally:

$$\text{BLEU} = \text{BP} \cdot \exp\left(\sum_{n=1}^{N} w_n \ln p_n\right), \quad \text{BP} = \exp\left(-\max\left(0, \frac{r}{c} - 1\right)\right),$$

$$p_n = \frac{\sum_{s \in G_n(\hat{y})} \min\left(C(s, \hat{y}),\ \max_{y \in \{y^{(i)}\}} C(s, y)\right)}{\sum_{s \in G_n(\hat{y})} C(s, \hat{y})},$$

where $G_n(\cdot)$ is the set of $n$-grams, $C(s, \cdot)$ is the count of $s$, $c$ is the candidate length, $r$ is the effective reference length, and $w_n$ are typically uniform weights over $n = 1, \ldots, 4$.

## B.2  ROUGE

ROUGE [24] (Recall-Oriented Understudy for Gisting Evaluation) measures the overlap of $n$-grams between a candidate and reference sentence. For ROUGE-N, recall, precision, and F1 are defined as:

$$\text{F1} = \frac{2 \cdot \text{P} \cdot \text{R}}{\text{P} + \text{R}}, \quad \text{R} = \frac{|\text{overlapping } n\text{-grams}|}{\text{total } n\text{-grams in reference}}, \quad \text{P} = \frac{|\text{overlapping } n\text{-grams}|}{\text{total } n\text{-grams in candidate}}.$$

Variants like ROUGE-L rely on the longest common subsequence (LCS).

## B.3  CIDEr

The CIDEr [38] (Consensus-Based Image Description Evaluation) metric evaluates the similarity of a generated caption to a set of human references by leveraging TF-IDF weighted $n$-gram statistics. Let $g_n$ and $r_n$ be the TF-IDF vectors for the candidate and reference captions, respectively. The score is computed as:

$$\text{CIDEr} = \frac{1}{N} \sum_{n=1}^{N} \frac{g_n \cdot r_n}{\|g_n\| \cdot \|r_n\|},$$

where $N$ denotes the number of $n$-gram orders considered (commonly $N = 4$). CIDEr rewards captions that align closely with human consensus while emphasizing informative, distinctive $n$-grams.

## B.4  BERTScore

BERTScore [47] evaluates the semantic similarity between candidate and reference captions by comparing contextual embeddings from pretrained language models such as BERT. Given embeddings $\mathbf{e}_c$ for candidate tokens and $\mathbf{e}_r$ for reference tokens, the precision, recall, and F1 are computed as:

$$F_1 = \frac{2 \cdot P \cdot R}{P + R}, \quad P = \frac{1}{|C|} \sum_{c \in C} \max_{r \in R} \cos\left(\mathbf{e}_c, \mathbf{e}_r\right), \quad R = \frac{1}{|R|} \sum_{r \in R} \max_{c \in C} \cos\left(\mathbf{e}_r, \mathbf{e}_c\right),$$

where $\cos(\cdot, \cdot)$ measures cosine similarity between token embeddings. Unlike BLEU and ROUGE, BERTScore captures semantic equivalence beyond exact $n$-gram overlaps and correlates better with human judgment.

### B.5 CLIPScore

CLIPScore [13] leverages pretrained CLIP (Contrastive Language–Image Pretraining) models to directly measure the alignment between a generated caption $\hat{y}$ and the corresponding image $I$. Unlike text-only metrics such as BLEU or ROUGE, CLIPScore evaluates captions in a multimodal embedding space, capturing whether the caption is semantically consistent with the visual content.

Formally, let $\mathbf{e}_I = f_{\text{img}}(I)$ and $\mathbf{e}_{\hat{y}} = f_{\text{text}}(\hat{y})$ be the normalized embeddings of the image and candidate caption from CLIP's image and text encoders. The score is computed as the cosine similarity:

$$\text{CLIPScore}(I, \hat{y}) = \cos\big(\mathbf{e}_I, \mathbf{e}_{\hat{y}}\big) = \frac{\mathbf{e}_I \cdot \mathbf{e}_{\hat{y}}}{\|\mathbf{e}_I\| \, \|\mathbf{e}_{\hat{y}}\|}.$$

Variants include length-penalized versions that discourage overly short captions and normalized versions that map scores into $[0, 1]$. CLIPScore has been shown to better correlate with human judgments of caption quality compared to traditional $n$-gram-based metrics, since it evaluates semantic faithfulness to the image rather than just textual overlap with references.

## C  Limitation

Currently, ViZer is limited to image captioning tasks, as extending the visual feature alignment to visual question answering (VQA) remains a challenging endeavor. In VQA, models often focus on localized regions or specific objects rather than the entire visual scene, making direct alignment between answers and global visual features a non-trivial task. In future work, we aim to design a scheme that enables ViZer to integrate VQA answer representations with corresponding visual semantics effectively.

Another limitation lies in the lack of suitable automated evaluation metrics for self-supervised VQA settings. Existing metrics either depend heavily on human-annotated references or rely on large language models, which may introduce additional biases. We plan to develop a reference-free, non-LLM-dependent scoring mechanism to more accurately evaluate alignment quality and enhance the reliability of performance assessment.

Finally, we envision integrating ViZer into a continuous learning loop with the underlying VLM. This would allow ViZer to evolve from an alignment module into a powerful pre-training strategy, enabling models to improve their multimodal grounding over time without requiring extensive manual supervision.

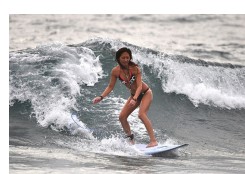

**GT:** A woman in a bikini riding a wave on a surfboard.
**Base:** <PERSON> in 2008
**RL:** <PERSON> in 2008
**ViZer$_{GT}$:** Woman surfing in the ocean.
**ViZer$_G$:** Surfer in a bikini riding a wave

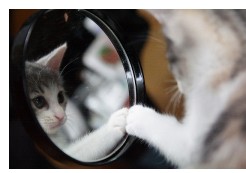

**GT:** A cat playing with it's reflection in a mirror.
**Base:** My cat is so cute.
**RL:** My cat has a mirror fetish.
**ViZer$_{GT}$:** A cat looking at herself in a mirror.
**ViZer$_G$:** A cat's paw in a mirror

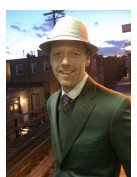

**GT:** a close up of a person wearing a suit and tie
**Base:** Me in 2010
**RL:** A man in a suit and hat on a train.
**ViZer$_{GT}$:** A man in a suit and hat on a train
**ViZer$_G$:** A man in a suit and hat on a train.

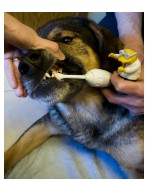

**GT:** A dog having his teeth brushed with a novelty electric toothbrush.
**Base:** A dog getting a dental cleaning.
**RL:** A dog getting a dental cleaning.
**ViZer$_{GT}$:** A dog getting his teeth brushed by his owner.
**ViZer$_G$:** A dog getting his teeth brushed with an electric toothbrush

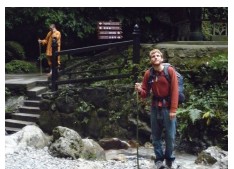

**GT:** A man holding a stick standing next to a green hillside.
**Base:** <PERSON> at the start of the trail
**RL:** <PERSON> and <PERSON> on the trail
**ViZer$_{GT}$:** A man with a backpack and trekking poles on the trail
**ViZer$_G$:** A man crosses a stream on the way to the temple.

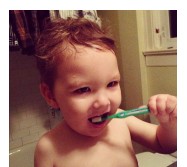

**GT:** A toddler in the bathroom brushing his teeth
**Base:** In this image we can see a child brushing his teeth.
**RL:** In this image we can see a child brushing his teeth with a toothbrush.
**ViZer$_{GT}$:** little boy brushing his teeth in the bathroom
**ViZer$_G$:** a little boy brushing his teeth in the bathroom

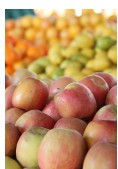

**GT:** A pile of apples sitting next to a pile of green apples.
**Base:** In this image I can see many fruits on the table.
**RL:** In this image I can see many fruits on the table.
**ViZer$_{GT}$:** In this image we can see a variety of fruits at a fruit stand
**ViZer$_G$:** In this image we can see a pile of fruits.

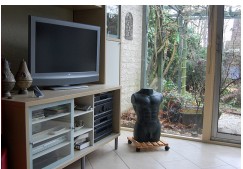

**GT:** A view of a TV and a statue laying on the ground next to a window.
**Base:** The living room in 2009
**RL:** The living room in 2009
**ViZer$_{GT}$:** A mannequin in front of a television set.
**ViZer$_G$:** A mannequin on a cart in the living room.

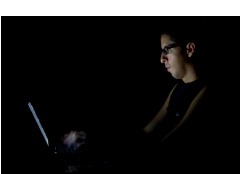

**GT:** A man with glasses sitting in front of a laptop computer.
**Base:** ITAP of a guy working in the dark
**RL:** ITAP of a guy working on his laptop in the dark
**ViZer$_{GT}$:** A man using a laptop in the dark.
**ViZer$_G$:** A man using a laptop computer in the dark

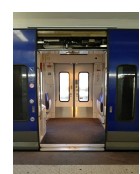

**GT:** The open doors looking into a subway from the station.
**Base:** Interior of a TGV
**RL:** Interior of a TGV
**ViZer$_{GT}$:** Interior of a train
**ViZer$_G$:** Interior of a train at the station

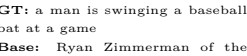

**GT:** a man is swinging a baseball bat at a game
**Base:** Ryan Zimmerman of the Washington Nationals
**RL:** Ryan Zimmerman of the Washington Nationals in 2007
**ViZer$_{GT}$:** Washington Nationals player batting during a game
**ViZer$_G$:** Washington Nationals' pitcher Stephen Strasburg (right) throws a pitch during the first inning of a game against the Washington Nationals at Nationals Park on April 19 2

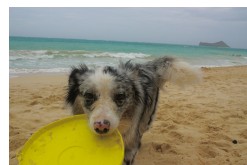

**GT:** There is a dog holding a Frisbee in its mouth.
**Base:** My doggo loves the beach!
**RL:** My doggo loves the beach!
**ViZer$_{GT}$:** My dog playing fetch on the beach.
**ViZer$_G$:** A pupper on the beach.

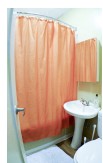

**GT:** A bathroom with a white toilet and a sink net to a shower.
**Base:** Bathroom with orange shower curtain.
**RL:** Bathroom with orange shower curtain.
**ViZer$_{GT}$:** A bathroom with a shower and sink.
**ViZer$_G$:** Bathroom with shower curtain and sink.

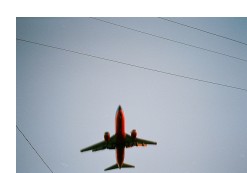

**GT:** A colorful plane flying over head and telephone wires
**Base:** ITAP of an airplane
**RL:** ITAP of an airplane
**ViZer$_{GT}$:** ITAP of an airplane flying in the sky
**ViZer$_G$:** ITAP of an airplane flying over power lines

Figure 5: Qualitative captioning comparison with SmolVLM-Base [29] for sample images from OpenImage [17] dataset. We compare ground-truth, baseline, reinforcement-learning, ViZer$_{GT}$, and ViZer$_G$ generated captions. Results continue on Figure 6

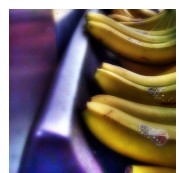

**GT:** A picture of a bunch of bananas sitting on a table.
**Base:** In this image we can see a bunch of bananas.
**RL:** In this image we can see a bunch of bananas.
**ViZer**$_{GT}$**:** Bunch of bananas on display at a grocery store.
**ViZer**$_G$**:** Bunch of bananas in a store

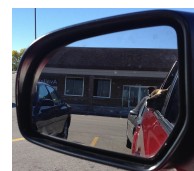

**GT:** rear view mirror reflection showing a dog in the back seat
**Base:** This dog is in my mirror
**RL:** This dog is in my mirror.
**ViZer**$_{GT}$**:** My dog in the rear view mirror.
**ViZer**$_G$**:** My dog is in the mirror.

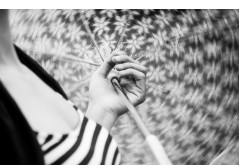

**GT:** A woman twirling a floral print parasol umbrella.
**Base:** ITAP of a girl holding an umbrella
**RL:** In this image I can see a person holding an umbrella.
**ViZer**$_{GT}$**:** A woman's hand holding an umbrella
**ViZer**$_G$**:** ITAP of a woman holding an umbrella

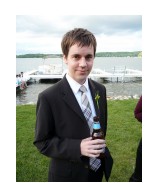

**GT:** A young man holding a bottle of beer while wearing a suit.
**Base:** Aaron in 2008
**RL:** Aaron in 2008
**ViZer**$_{GT}$**:** A man wearing a suit and tie holding a beer.
**ViZer**$_G$**:** Aaron in his wedding suit.

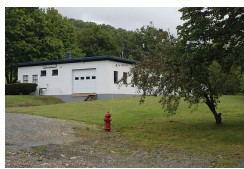

**GT:** A red fire hydrant sitting in a field next to a white building.
**Base:** The firehouse in 2014
**RL:** The firehouse in 2014
**ViZer**$_{GT}$**:** The fire station in 2009
**ViZer**$_G$**:** The fire station in 2013

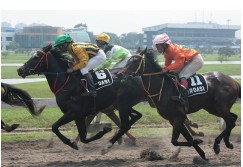

**GT:** Horses and their riders are racing in a horse race.
**Base:** Horse racing at the 2010 Asian Games
**RL:** Horse racing at the 2010 Asian Games in Jakarta
**ViZer**$_{GT}$**:** Jockeys on horses during a race.
**ViZer**$_G$**:** Horse racing at the 2010 Hong Kong Horse Racing Festival

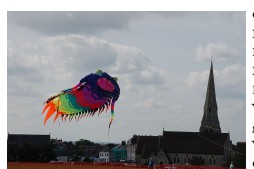

**GT:** A kite is flying in a clear sky.
**Base:** Kite at the 2013 Bristol Kite Festival
**RL:** Kite at the 2013 Bristol Kite Festival
**ViZer**$_{GT}$**:** Kite flying in the background of a church steeple.
**ViZer**$_G$**:** Kite flying at the 2012 Great Yarmouth Kite Festival

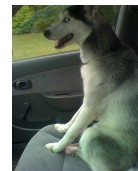

**GT:** A grey and white dog sitting in the passenger side of a car.
**Base:** My husky is a princess.
**RL:** My husky is a princess.
**ViZer**$_{GT}$**:** My husky sitting in the back seat
**ViZer**$_G$**:** This is my dog. She's a husky.

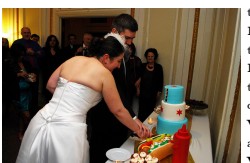

**GT:** A newly married couple cutting up a giant hot dog cake.
**Base:** A bride and groom cutting their wedding cake at a reception.
**RL:** A bride and groom cutting their wedding cake at a wedding reception.
**ViZer**$_{GT}$**:** Couple cutting the cake at a wedding reception
**ViZer**$_G$**:** Couple cutting the wedding cake at the reception

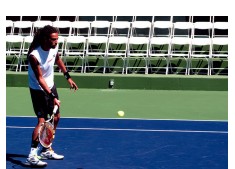

**GT:** a man playing tennis going for a low ball
**Base:** Djokovic at the 2008 US Open
**RL:** Serena Williams at the 2008 U.S. Open
**ViZer**$_{GT}$**:** Tennis player hitting a ball during a practice session.
**ViZer**$_G$**:** Tennis player Rafael Nadal warming up before a match at the 2008 Australian Open.

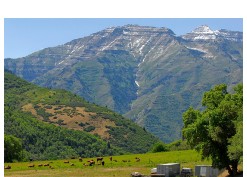

**GT:** Cows lounge in a field with a mountain backdrop.
**Base:** Mt. Timpanogos in the background
**RL:** Mt. Timpanogos in the background
**ViZer**$_{GT}$**:** A view of the mountains in the background with cattle grazing in a field.
**ViZer**$_G$**:** Cows grazing in the valley below the mountains.

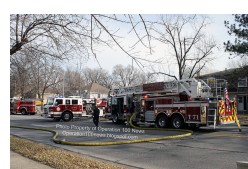

**GT:** A fire fire truck parked in a parking lot.
**Base:** Photo Property of Operation 100 News
**RL:** Photo Property of Operation 100 News
**ViZer**$_{GT}$**:** Firefighters with a ladder truck in front of a house on fire.
**ViZer**$_G$**:** Firefighters at the scene of a fire in the early morning light.

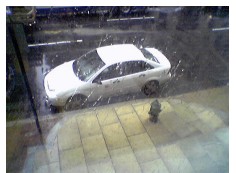

**GT:** A car is parked on the side of the street in the rain.
**Base:** The bus stop in front of my office
**RL:** The bus stop in front of my office.
**ViZer**$_{GT}$**:** The view from my office window.
**ViZer**$_G$**:** A taxi cab parked in front of the station

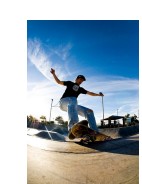

**GT:** A man on a skateboard is coming up a ramp.
**Base:** In this image we can see a person riding a skateboard.
**RL:** In this image we can see a person riding a skateboard.
**ViZer**$_{GT}$**:** In this image we can see a skateboarder
**ViZer**$_G$**:** In this image we can see a person riding a skateboard on a ramp.

Figure 6: Continuation of Figure 5. Generated caption samples for the OpenImage dataset.

**GT:** a few people that are standing on a beach
**Base:** The image shows a group of people on a beach, with some standing and others sitting on the sand. There is a surfboard lying on the sand
**RL:** The image shows a group of people on a sandy beach, with some standing and others sitting or lying down. There is a surfboard lying on the sand, and a red flag is visible in the
**ViZer$_{GT}$:** The image depicts a group of people enjoying a day at the beach. There are several individuals standing and sitting on the sand, some of whom are holding surfboards. The beach is lined with palm trees
**ViZer$_G$:** The image depicts a group of people enjoying a day at the beach. They are standing and sitting on the sand, with some of them holding surfboards. The beach is lined with palm trees and there

**GT:** Cows lounge in a field with a mountain backdrop.
**Base:** The image depicts a mountainous landscape with a group of cows grazing in a meadow below. The mountains are covered in greenery and have patches of
**RL:** The image depicts a mountainous landscape with a group of cows grazing in a meadow below. The mountains are covered in greenery and have patches of snow on their peaks. The sky is clear,
**ViZer$_{GT}$:** The image depicts a scenic mountainous landscape with a herd of cows grazing in a meadow. The mountains are covered in lush greenery and have snow-capped peaks, creating a stunning natural backdrop.
**ViZer$_G$:** The image depicts a scenic mountainous landscape with a group of cows grazing in a meadow. The mountains are covered with lush green vegetation, and the sky is clear with a few scattered clouds. The

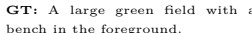
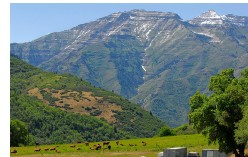
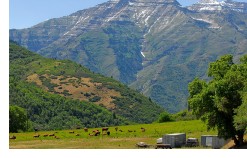

**GT:** A large green field with a bench in the foreground.
**Base:** The image depicts a park bench situated on a grassy hillside. The bench is made of wood and has a blue and red painted backrest.
**RL:** The image depicts a park bench situated on a grassy hillside. The bench is made of wood and has a blue and red painted backrest. The surrounding area is lush with green grass, and
**ViZer$_{GT}$:** The image depicts a serene park setting with a well-maintained grassy hill. The hill is covered with lush green grass, and there are several trees scattered throughout the area. In the foreground,
**ViZer$_G$:** The image depicts a serene park setting with a well-maintained grassy hill. On the left side of the image, there is a wooden bench with a colorful, weathered paint job. The

**GT:** A car is parked on the side of the street in the rain.
**Base:** The image shows a white car parked on the side of a street during a rainy day. The scene is captured through a window, and the raindrops
**RL:** The image shows a white car parked on the side of a street during a rainy day. The scene is captured through a window, and the raindrops are visible on the glass. There is also a
**ViZer$_{GT}$:** The image shows a white sedan car parked on a wet street during heavy rain. The car's headlights are on, and the windshield wipers are in motion, indicating that it is raining. There is
**ViZer$_G$:** The image shows a white car parked on the side of a street during a rainy day. The car appears to be stationary, and there is a puddle of water on the ground in front of it

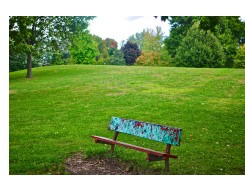
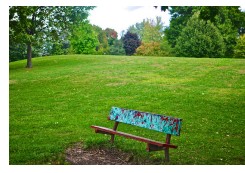

**GT:** a yellow and green train, at stop lights.
**Base:** The image shows a green and yellow train traveling on tracks. The train is passing under a large metal structure, which appears to be part of a railway
**RL:** The image shows a green and yellow train traveling on tracks. The train is passing under a large metal structure, which appears to be part of a bridge or overpass. The sky is cloudy, and
**ViZer$_{GT}$:** The image shows a green and yellow train traveling on tracks. The train is passing under a large metal structure, which appears to be part of a bridge or overpass. The sky is cloudy, and
**ViZer$_G$:** The image features a green and yellow train with a yellow circular logo on its side. The train is moving on tracks, and there is a large metal structure in the background, possibly part of a railway

**GT:** A dog rests his head on the edge of a boat at sea.
**Base:** The image shows a German Shepherd dog sitting on the deck of a boat. The dog is wearing a colorful collar with a heart-shaped tag. The background
**RL:** The image shows a German Shepherd dog sitting on a boat, looking out over the water. The dog is wearing a colorful collar with a heart-shaped tag. The background features a clear blue sky and a
**ViZer$_{GT}$:** The image shows a German Shepherd dog sitting on the deck of a boat. The dog is wearing a colorful collar with a heart-shaped tag, and it appears to be looking out at the water. The
**ViZer$_G$:** The image shows a German Shepherd dog sitting on a boat, with its head resting on a railing. The dog is wearing a colorful collar with a heart-shaped tag. The background features a scenic view of

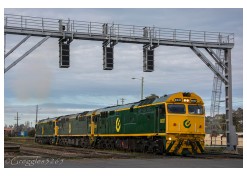
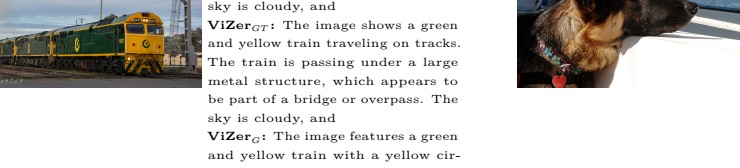

Figure 7: Qualitative captioning comparison with Qwen2-VL [39] for sample images from OpenImage [17] dataset. We compare ground-truth, baseline, reinforcement-learning, ViZer$_{GT}$, and ViZer$_G$ generated captions. Results continued in Figure 8

**GT:** A man with glasses sitting in front of a laptop computer.

**Base:** The image shows a person sitting in a dark room, working on a laptop computer. The individual is wearing glasses and appears to be focused on the screen

**RL:** The image shows a person sitting in a dark room, working on a laptop computer. The individual is wearing glasses and appears to be focused on the screen. The lighting is low, creating a dramatic and

**ViZer$_{GT}$:** The image shows a person sitting in a dimly lit room, working on a laptop computer. The screen of the laptop is illuminated, and the person appears to be focused on their work. The overall

**ViZer$_G$:** The image shows a person sitting in a dark room, working on a laptop. The individual is wearing glasses and appears to be focused on the screen. The lighting is low, creating a dramatic and intense

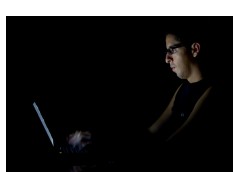

**GT:** a white boat some green hills and water

**Base:** The image depicts a small boat sailing on calm waters near a mountainous coastline. The boat has a traditional design with a mast and sails, and it

**RL:** The image depicts a small boat sailing on calm waters near a mountainous coastline. The boat has a traditional design with a mast and sails, and it appears to be a fishing or leisure vessel. The

**ViZer$_{GT}$:** The image depicts a small sailboat floating on calm waters near a mountainous landscape. The boat has a traditional design with a mast and sails, and it appears to be a fishing vessel. The surrounding

**ViZer$_G$:** The image depicts a small boat floating on calm water, surrounded by a body of water that appears to be a lake or a sea. The boat has a white hull with some orange and yellow accents.

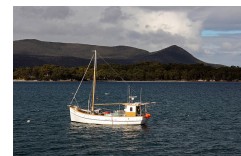

**GT:** A crowded kitchen that is cluttered and messy.

**Base:** The image shows a small kitchen with white cabinets and appliances. There is a refrigerator, a stove, and an aquarium on the counter. The kitchen also

**RL:** The image shows a small kitchen with white cabinets and appliances. There is a refrigerator, an oven, and a sink visible. The kitchen also has a fish tank on the countertop and some decorative items

**ViZer$_{GT}$:** The image shows a kitchen with various appliances and furniture. There is a refrigerator, a stove, a sink, and an aquarium. The kitchen also has cabinets, drawers, and a window with curtains.

**ViZer$_G$:** The image shows a small, cluttered kitchen with various appliances and furniture. The kitchen features a white refrigerator, a stove with a black oven, and a sink. There is a cabinet above the sink

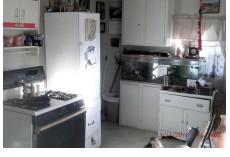

**GT:** A man stands poised with a baseball bat.

**Base:** The image is a vintage photograph of a baseball player in action. The player is holding a baseball bat and appears to be in the middle of a swing

**RL:** The image is a vintage photograph of a baseball player in action. The player is holding a baseball bat and appears to be in the middle of a swing. He is wearing a dark uniform, including a

**ViZer$_{GT}$:** The image is a black-and-white photograph of a baseball player from the late 19th or early 20th century. The player is holding a baseball bat and appears to be in the

**ViZer$_G$:** The image is a sepia-toned photograph featuring a vintage baseball player in mid-swing. The player is dressed in a dark uniform, including a cap, and is holding a baseball bat. The

**GT:** A little girl that is standing with an umbrella.

**Base:** In the image, a young girl is running down a narrow, gravel-covered street. She is wearing a pink dress and has a pink umbrella in her

**RL:** In the image, a young girl is running down a narrow, gravel-covered street. She is wearing a pink dress and has a pink umbrella in her hand. The street is flanked by green plants

**ViZer$_{GT}$:** In the image, a young girl is running down a narrow, unpaved street. She is wearing a pink dress and has a pink umbrella in her hand. The street is lined with green plants and

**ViZer$_G$:** In the image, there is a young girl wearing a pink dress and holding a pink umbrella. She appears to be walking down a narrow, unpaved street. On the right side of the street,

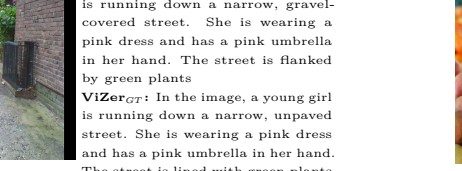

**GT:** A pile of apples sitting next to a pile of green apples.

**Base:** The image shows a variety of fruits, including apples and oranges. The apples are red and yellow, while the oranges are a mix of orange and yellow

**RL:** The image shows a variety of fruits, including apples and oranges. The apples are red and yellow, while the oranges are a mix of orange and yellow hues. The fruits are arranged in a pile,

**ViZer$_{GT}$:** The image shows a close-up view of a variety of fruits, including apples and oranges. The apples have a mix of red and yellow hues, while the oranges are a vibrant orange color. The fruits

**ViZer$_G$:** The image shows a close-up view of a variety of fruits, primarily apples and oranges. The apples are predominantly red and yellow, with some green ones interspersed. The oranges are a mix of

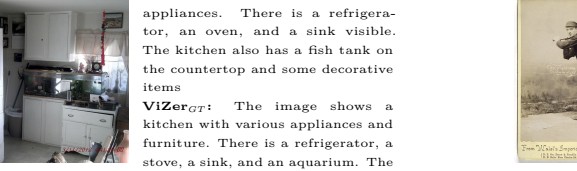
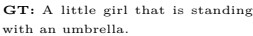
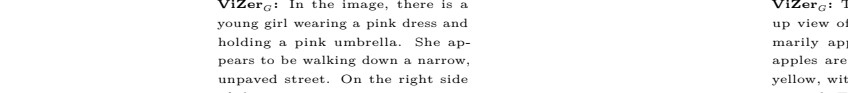

Figure 8: Continuation of Figure 7. Generated caption samples for the OpenImage dataset.