# OpenReview forum: "Unifying Vision-Language Latents for Zero-label Image Caption Enhancement"
_NeurIPS.cc/2025/Workshop/UniReps — UniReps2025_

### Official Review · Reviewer_tvEz · 2025-09-03
**Good take on text/image latent alignment at training time**

**Confidence:** 4

**Review:**

I enjoyed reading this paper. The central idea of actively aligning vision and language latent spaces to enhance zero-label image captioning is well-motivated and clear. The authors do a good job of positioning their approach in the broader landscape of vision-language research, and the experiments, while compact, are convincing enough to demonstrate the method's promise. I appreciated the emphasis on qualitative results; the examples help highlight how ViZer tends to produce captions that are more grounded and descriptive, even when quantitative metrics fail to reflect these gains.

I also quite appreciated the final discussion. The observation that (quote) "a central challenge lies in the inadequacy of automated evaluation metrics for image captioning" is quite timely. This is a subtle but important point: our current metrics, whether reference-based or reference-free, often misrepresent the true quality of generated captions, especially in scenarios involving zero-label learning. Bringing this issue to the foreground positions the paper within a broader research conversation that will resonate with the community.

That said, I was a bit surprised not to see a discussion of two related lines of work. The first is "LiT: Zero-Shot Transfer With Locked-Image Text Tuning", which established a precedent for using a fixed encoder effectively. The second (and in my opinion more relevant) is "ASIF: Coupled Data Turns Unimodal Models to Multimodal without Training". In the latter paper, two frozen encoders can be aligned without additional training or layers, simply via a relative intermediate representation. Such a representation could have served as a lightweight intermediate step in ViZer's pipeline, potentially reducing the need for mapper training or improving initialization. At minimum, a brief discussion of how ViZer relates to or diverges from LiT and ASIF would strengthen the paper and help readers position it in the evolving literature.

A few other points could be tightened or clarified. The paper emphasizes the modularity and efficiency of ViZer, but it would help to discuss the computational overhead in more concrete terms. Similarly, while the qualitative improvements are clear, the quantitative results (though understandably limited by the metrics) might benefit from additional ablation studies or error analysis to tease out where ViZer succeeds or struggles most. Finally, it might be useful to discuss how the framework could scale to larger VLMs or more diverse domains, as this seems like a natural next step.

Overall, this is a well-executed research that makes a meaningful contribution to the study of zero-label multimodal learning. I believe it is a strong fit for UniReps.

**Score:**

4

**Topic Fit:**

3

---

### Official Review · Reviewer_Rz9b · 2025-09-10
**Unifying Vision-Language Latents for Zero-label Image Caption Enhancement**

**Confidence:** 4

**Review:**

This paper proposes an approach to enhance image caption generation by re-aligning image and VLM representations in the latent space during post-pretraining. The method addresses the lack of ground-truth image captions for training and is evaluated through both qualitative and quantitative experiments.

Pros:
1) Aligning image and VLM representations in the latent space for caption generation is a creative and relevant direction.
2) The qualitative results are compelling and help contextualize the metric-based evaluations.
3) The paper is generally well written, with only minor issues in presentation.

Cons:
1) The phrase “images that the model has already seen during previous training steps” is unclear. Does this refer to pretraining or the ViZer_GT training stage? More precision is needed. If such image data is reused, why not compare against direct finetuning with it? A direct finetuning baseline seems important, especially since only an RL-based method is compared.
2) The paper would benefit from schematic illustrations or flow diagrams of ViZer_GT and ViZer_G (Figure 2 only depicts ViZer_G). This would make their differences more immediately clear to the reader.
3) Since ViZer_GT uses ground-truth information, it should be compared fairly against a direct finetuning method. Otherwise, it risks overstating the strength of ViZer_GT relative to realistic baselines. Also, LoRA is used in this work but it'd be useful that both full model training and lora results are presented.
4) Figures 5–8 do not highlight the key differences between methods. Most examples favor the proposed approach, but the reader also needs to see cases where it underperforms (e.g., hallucinations or failure cases). This would give a more balanced view of strengths and weaknesses.

The paper presents a promising and well-motivated idea, but several clarifications and additional experiments are needed to make the evaluation more convincing. In particular, comparisons against direct finetuning, clearer visual flows for the proposed mappers, and explicit analysis of failure cases would significantly strengthen the work. Also, this work seems to be relevant: https://arxiv.org/abs/2504.12717

**Score:**

2

**Topic Fit:**

2

---

### Official Review · Reviewer_upTP · 2025-09-14
**Review of Submission 95**

**Confidence:** 4

**Review:**

The paper proposed a framework for training image captioning models without the need for large amounts of labeled data. It does this by refining a cross-modal alignment of vision and language representations, during training. The approach is evaluated on two existing vision-language models, and compared with a baseline using a number of metrics. More emphasis is ultimately placed on qualitative comparisons, and it is found that the proposed method leads to more grounded and descriptive captions.

Strengths:
 - The approach leverages recent advances in unified semantic representation across modalities, thus reducing the reliance on large labeled datasets.
 - Quantitative evaluations are extensive, and unpacked with care and insight.

Weaknesses:
 - A minor point: In the experiments section, the discussion somewhat unexpectedly switches to a critique on existing automated evaluation metrics for caption generation. While good and valid points are raised, it did not seem to be a central aspect of the paper up to that point.

**Score:**

3

**Topic Fit:**

2

---

### Official Review · Reviewer_AHzS · 2025-09-15
**The work proposes an approach that could align the visual and text features when there is no high quality label available for the textual modality specifically. For this purpose they propose a network Vizer, which is a lightweight “mapper” that aligns vision and language latent spaces using a cosine-similarity objective and then the authors train a small VLM for image captioning without using caption labels during VLM training.**

**Confidence:** 3

**Review:**

Strengths: The work has several strengths as listed below:

1. A lightweight mapper aligns latent spaces with a simple cosine similarity loss; it can be viewed as a post-hoc between a frozen vision encoder and an LLM for easy fine tuning and adaptation.

2. I really admired the honest discussion in the paper about the limitations of the caption metrics and why reference-free metrics still fall short, plus qualitative analyses/visualizations.

Weakness: The paper has the following weakness, which basically affected the overall ranking:

1. The "zero-label" claim is not completely justified. While the final VLM is trained without text labels, ViZerGT uses labeled COCO/CC3M to train the particular mapper; only ViZerG is truly zero-label wrt captions. So, the entire claim that this is zero-label training is not valid, as the pre-training of developing the lightweight mapper still utilizes the text labels.

2. Lack of baselines and ablations: (i) The RL baseline is included but even the paper says the comparison is unfair (RL sees ground-truth captions). Other related baselines are missing, for instance, SFT with synthetic captions and stronger instruction-tuned captioners. The authors need to show these results to really make a good justification for their approach; (ii) The paper lack several ablations: comparisons of different alignment losses (e.g., InfoNCE, MSE) were not even tried, only sticked to cosine-similarity loss, where to insert the mapper in the overall architecture, there was no intuition or motivation beghind this and not tried to explore other positions.

**Score:**

2

**Topic Fit:**

2